# Fusion of Dense Airborne LiDAR and Multispectral Sentinel-2 and Pleiades Satellite Imagery for Mapping Riparian Forest Species Biodiversity at Tree Level

**DOI:** 10.3390/s24061753

**Published:** 2024-03-08

**Authors:** Houssem Njimi, Nesrine Chehata, Frédéric Revers

**Affiliations:** 1UR17DN01, Aviation School of Borj El Amri, Borj El Amri 1142, Tunisia; 2UMR EPOC, Bordeaux INP, 33400 Talence, France; 3BIOGECO, INRAE, University Bordeaux, F-33615 Pessac, France; frederic.revers@inrae.fr

**Keywords:** multispectral, Sentinel-2, Pleiades, LiDAR, data fusion, forest biodiversity, species classification

## Abstract

Multispectral and 3D LiDAR remote sensing data sources are valuable tools for characterizing the 3D vegetation structure and thus understanding the relationship between forest structure, biodiversity, and microclimate. This study focuses on mapping riparian forest species in the canopy strata using a fusion of Airborne LiDAR data and multispectral multi-source and multi-resolution satellite imagery: Sentinel-2 and Pleiades at tree level. The idea is to assess the contribution of each data source in the tree species classification at the considered level. The data fusion was processed at the feature level and the decision level. At the feature level, LiDAR 2D attributes were derived and combined with multispectral imagery vegetation indices. At the decision level, LiDAR data were used for 3D tree crown delimitation, providing unique trees or groups of trees. The segmented tree crowns were used as a support for an object-based species classification at tree level. Data augmentation techniques were used to improve the training process, and classification was carried out with a random forest classifier. The workflow was entirely automated using a Python script, which allowed the assessment of four different fusion configurations. The best results were obtained by the fusion of Sentinel-2 time series and LiDAR data with a kappa of 0.66, thanks to red edge-based indices that better discriminate vegetation species and the temporal resolution of Sentinel-2 images that allows monitoring the phenological stages, helping to discriminate the species.

## 1. Introduction

Forests, which cover around one-third of continental surfaces [1,2], constitute a source of materials and renewable energy; they also provide major ecosystemic services such as soil preservation, biodiversity conservation and climate regulation [3]. In order to ensure sustainable management of forest resources, the study of their functioning and dynamics is essential. Multispectral and 3D LiDAR remote sensing data sources have proven to be valuable tools for modeling forest structure [4,5,6,7,8] and detecting vegetation strata [9], thus helping to understand the relationship between forest structure, tree species diversity, and microclimate.

The global objective of this work is to produce relevant information from multi-source LiDAR and optical multispectral data such as spectral indices, Digital Terrain Models (DTM), species classification, 3D structures, and vegetation profiles in order to link them to tree biodiversity indicators. This study mainly focuses on mapping riparian forest species in the canopy strata using a fusion of Airborne LiDAR data and multispectral multi-source satellite imagery: Sentinel-2 and Very High Resolution (VHR) Pleiades at tree level.

Previous studies [4,10,11] revealed that Sentinel-2’s narrow bands located in the red edge (B6, B7 and B 8a) and indices derived from them, such as NDVIre [5], help to overcome the well-known problem of saturation of the vegetation. This has provided support for the use of such spectral indices in the forest species classification to conquer the complex spatial properties (complexity of the forest environment, variability of crown diameters, overlaps of vegetation) and very similar radiometric properties (shades of green and shadows) as well as the spatial resolution that is not sufficient to discern species with the bare eye. In addition, some other studies [6,12] have shown that the results of multi-date classification based on seasonal analysis [7] surpassed those of single-date classification, pointing out the importance of Sentinel-2 temporal resolution for mapping forests.

As for VHR imagery, their advantage is exploiting geometric information using oriented object methods thanks to segmentation algorithms. Individual tree crown delineation algorithms based on 2.5 D Canopy Height Models (CHMs), such as itcSegment [8], SEGMA [13], and eCognition [14], are faster than those based on 3D point clouds (AMS3D [15], Graph-Cut [16,17], Profiler [18,19]). Conversely, 3D algorithms showed better crown delineation results than counting on CHM only, especially with dense 3D point clouds [14]. 

The most innovative idea of this work is the establishment of multi-level data fusion methodology for mapping riparian forests at tree level in a fully automated scheme developed by means of the Python programming language. It consists of data co-registration, feature fusion, and finally, decision-making fusion stages. The contribution of each data source in the mapping process was assessed by testing four data fusion configurations. This paper provides the study findings through the following structure: Materials and Methods are presented in Section 2, Results and Discussion are respectively given in Section 3 and Section 4, and finally, conclusions are drawn in the Section 5. 

## 2. Materials and Methods

### 2.1. Study Area

The study area is located in southwestern France (Figure 1). The Ciron watershed and its riparian forest is a tributary of the Garonne, known as a climatic refuge for the beech, on the warm margin of its European range [20]. This riparian forest is made up of an assemblage of species such as oak, beech, locust, pine, etc. Twenty-eight plots forming a gradient of three-dimensional vegetation structure are defined. They are distributed along a 30 km stretch of the Ciron and along a 5 km tributary of the Ciron in which the riparian forest is lined with pine forests (maritime pine) in order to homogenize the potential impact of the surrounding landscape on the biodiversity of fauna and flora in the riverine.

### 2.2. Remote Sensing Data

In order to take advantage of the complementarity between the spectral information resulting from optical data and geometric information from LiDAR data for the characterization of forest species, three types of multi-source and multi-resolution data were processed. They are based on Sentinel-2 image time series, Pleiades VHR images, and LiDAR 3D point clouds.

#### 2.2.1. Satellite Images

##### Sentinel-2 Images

The Sentinel-2 mission is a high-resolution optical mission of the European Space Agency (ESA). It is a constellation of two identical satellites, Sentinel-2A and Sentinel-2B, for Earth observation and launched since 2015.

French public institutions involved in Earth observation and environmental sciences created the Theia continental surface data and services hub [21]. This center provides the international scientific community with a panoply of satellite images, including Sentinel-2 images with different levels of pre-processing.

The Sentinel-2 sensor provides 10 spectral bands dedicated to earth observation at 10 m and 20 m, covering Vis-NIR, red edge, and SWIR domains, which are useful for vegetation classification. In addition, Sentinel-2 A and B allow us to have a free and easy downloadable time series with a 5-day frequency, allowing us to perform classifications on multi-temporal image series.

A collection of 11 Sentinel-2 satellite images was used, ranging from January to December 2019, and processed in level 2A, thus providing top-of-canopy reflectance and reducing the effects of slopes and shadows. Information about the scenes used is provided in Table 1, and a sample of a Sentinel-2 image covering site 1 is provided in Figure 2a.

##### VHR Pleiades Images

Pleiades is an environment-focused constellation consisting of two satellites (referred to as 1A and 1B) from CNES (French center of spatial studies) that were launched on 17 December 2011 (Pleiades 1A) and 2 December 2012 (Pleiades 1B). It is characterized by a very high spatial resolution of 50 cm for the panchromatic band and 2 m for multi-spectral bands, as well as four spectral bands B, G, R, and NIR and a temporal resolution of 26 days [19]. Pleiades images were processed in top-of-the-atmosphere reflectance (TOA). Figure 2b shows the Pleiades image covering site 01’s extent.

#### 2.2.2. LiDAR Data

LiDAR is the only remote sensing technology that allows the user to get 3D information underneath the vegetation canopy and to model the 3D vegetation structure since the laser impulsion penetrates the vegetation. In this work, a 250 m vertical height flight acquisition mission with a 190 kHz measurement rate provided a very dense point cloud of 68 pts/m^2^. This helps with modeling the soil underneath vegetation and the 3D vegetation structure. Table 2 and Figure 2c, respectively, summarize the LiDAR mission details and show a sample of the LiDAR point cloud clipped on site 01’s extent.

#### 2.2.3. Ground Truth Data

Joint airborne acquisition and in-field observations were conducted in the autumn (3 and 4 October 2019) with tree foliage on. All the measurements were carried out on each site within a 15 m radius plot (2229 trees in total), including all the riparian forest, as well as a part of the pine forest adjacent to the riparian forest. Differential GNSS (DGNSS, Trimble, Westminster, CO, USA) was used to measure the plot center coordinates. For all the trees with a diameter at breast height (DBH) above 7.5 cm, trunk circumferences at breast height were measured with a tape, and tree heights were measured using a hypsometer Vertex (Haglöf Sweden, Långsele, Sweden). The data processing then required a delimitation of the 28 sites. The sites were chosen to represent a gradient in the width and density of the riparian forest. Table 3 summarizes some field plot measurements [22].

Square-shaped vector layers with a side of 200 m each were used to crop optical and LiDAR data. These layers were projected into the “RGF 93 Lambert 93” coordinate system. In-field work led to more than 31 unbalanced classes. However, in this study, we only focused on five major canopy classes. Training and Testing data (Figure 3) were selected using individual tree crowns generated after the segmentation process (see Section 2.3.3). They consist of 165 and 73 samples, respectively.

### 2.3. Methodology

In this paper, an automated workflow is proposed for airborne LiDAR and multispectral satellite imagery fusion to map riparian forest species biodiversity at tree level, as illustrated in the scheme below (Figure 4).

It consists of three levels of processing: LiDAR and multispectral data were first co-registered, exploiting the high accuracy of LiDAR data. Each data source was processed separately and then fused at the feature and decision-making levels [6].

#### 2.3.1. LiDAR and Multispectral Images Co-Registration

Co-registration is essential to manipulate multi-source data. However, the digital elevation models used for satellite image ortho-rectification are less accurate than those generated from LiDAR point clouds due to the differences between their spatial resolutions. This leads to a misregistration between the image and the LiDAR-derived elevation models. We, therefore, opted for a data co-registration as a geometric correction that consists of translation of spectral bands in order to assign them the same planimetric location as the 2D representations of LiDAR data and match their pixels with accurate tree positions.

In the context of species characterization in a forest environment, the most widely used 2D representation is the Canopy Height Model (CHM), as it represents the height of trees. It is obtained by subtracting the Digital Terrain Model from the Digital Surface Model. However, DSM, DTM, and CHM (Figure 5) were derived from dense LiDAR point clouds, leading to 2D elevation maps at a resolution of 0.25 m. Spectral bands were then upsampled to the CHM resolution using nearest neighbor interpolation and clipped over the plots’ extents.

Field data and tree measurements had a major role in the co-registration process, as the trees’ geographical coordinates and measured heights helped with pointing out the trees in the most geometrically accurate data source, i.e., LiDAR data (CHM). On the contrary, Sentinel-2 imagery suffers from a degraded spatial resolution (10 m) and an absolute geolocation accuracy (CE95) of 8 m [23]. The points corresponding to in-field trees were pointed in the image pixel centers. Finally, a 1st polynomial transformation was applied for the co-registration process. Figure 6 shows the translation step of co-registration of a spectral band and a 2D representation of LiDAR data CHM (base layer).

#### 2.3.2. Feature Fusion

Fusion at the feature level was first processed by generating and then stacking spectral and geometric LiDAR 2D attributes, which are intended to be used in the classification of forest species. We mainly used three LiDAR attributes, as shown in Table 4 and Figure 7.

Seven vegetation indices [3] were derived for each date based on near-infrared and red-edge channels: NDVI, GRVI1, CIre, NDVIre3, NDre2, SAVI, MSAVI2 [24]. Soil-adjusted indices such as SAVI and MSAVI2 were used to better handle non-dense tree species (Table 5).

#### 2.3.3. Decision-Making Fusion

At the decision-making level, 3D LiDAR point clouds were segmented using the PyCrown method [25], which is a re-implementation of the itcSegment crown delimitation algorithms. It provides a 3D segmentation of individual trees besides a raster segmentation. It is based on local maximum search (i.e., treetops) and region growth with regard to user-defined parameters (distance of a crown point from its top and point height with respect to crown average heights). These parameters are defined for each site individually with the aim of maximizing the compactness [26].

Segmented LiDAR regions were first used to select training and testing samples by inspecting the ground truth data and assigning the appropriate classes to tree crowns. Then, spectral attributes at an object level (tree crowns) were derived using the attributes’ mean and standard deviation over each segmented region. An object-oriented classification was then carried out on labeled tree crowns. This is where the decision-making fusion between LiDAR and multispectral data lies.

Due to few training data, data augmentation techniques such as Gaussian Noise filtering were applied on each spectral band and each attribute to double them in order to allow the classifier to learn more robust features [27].

Different feature combinations were used to assess the importance of spectral, temporal, or spatial information:Single date Sentinel-2 and LiDAR fusionSentinel-2 time series and LiDAR fusionPleiades and LiDAR fusionSingle date Sentinel-2, Pleiades, and LiDAR fusion

Finally, the classification was processed using a random forest classifier. Results were evaluated using overall accuracy, kappa, and per-class precision and recall.

#### 2.3.4. Data Fusion Process Automation

A Python script was developed for automating data fusion for tree species classification using GDAL, otbApplication, Numpy, Whitebox, and PyCrown (Table 6).

## 3. Results

### 3.1. Tree Crown Delineation

The individual tree crown delineation process provided good results in both 2D and 3D segmentation, as shown in Figure 8 and Figure 9.

### 3.2. Classification

Object-based classification was carried out on individual trees generated previously. A random forest algorithm was applied for classification using 200 trees. Gini measure was used for splitting the nodes using $\sqrt{M}$ random features, with *M* the number of input features. Table 7 summarizes the overall classification accuracies with different fusion configurations and measures data augmentation impact, which improved kappa and overall accuracy by 15% and 8%, respectively. Table 8 presents the precision and recall values per species for Sentinel-2 time series and LiDAR fusion with data augmentation, which had the best classification results.

Some classification maps for different sites are given in Figure 10, Figure 11 and Figure 12.

## 4. Discussion

### 4.1. Quantitative Interpretations

#### 4.1.1. Evaluation of the Best Fusion Configuration

The best classification results were obtained by the combination of multi-temporal Sentinel-2 images and LiDAR data. We obtained 0.66 as kappa and 0.69 as overall accuracy. It was also shown that Tauzin Oak, Maritime Pine, and the class Other have precision values superior to 0.73, which means that at least 73% of these species were correctly labeled. Otherwise, less than 54% of Pedunculate Oak and Black Alder were correctly labeled. Results have also shown low recall values for the class Other compared to other classes, which means that it is sub-estimated by the classifier. In fact, this class is heterogeneous, as it assembles many minority tree species that do not essentially have similar characteristics. Thus, confusion occurred in the learning process, causing lower classification results.

#### 4.1.2. Contribution of Data Augmentation

The results obtained by increasing training and validation samples are more relevant. Indeed, this allowed us to have more training data to better optimize the classification model parameters. Moreover, the initial training and testing polygons respectively correspond to 0.35% and 0.15% only of the total number of tree crowns. The small number of polygons selected is mainly related to the fact that the in-field data provided are point-geometry shapefiles (Figure 13) covering only 15 m radius circular plots. Each point corresponds to a location of a tree trunk. The density of these points is important so that some canopy crowns could cover sub-canopy trees. This makes the choice of polygons critical in order to avoid the interference of several classes within the same polygon and consequently avoid misleading the learning process. Hence, the number of training and testing samples was reduced.

### 4.2. Qualitative Interpretations

#### 4.2.1. Spectral vs. Spatial Resolution Contribution

By comparing the results of “LiDAR—Sentinel-2 mono-date” and “LiDAR—Pleiades” fusion configurations, one can notice that the first combination using Sentinel-2 imagery was better despite a lower spatial resolution than Pleiades imagery (2 m). This can be explained by the fact that the forestry environment is dominated by vegetation, leading to low reflectance variance in visible and near-infrared spectral bands of Pleiades. In addition, the higher the spatial resolution is, the more the satellite sensor is able to discern small details. Therefore, shadows and ground through the vegetation are detected, which makes discriminating species more challenging.

However, Sentinel-2 images are characterized by spectral richness (10 bands). Thanks to its high spectral resolution, and especially the red-edge domain, many attributes are calculated that highlight information related to chlorophyll activity, chlorophyll content, internal leaf structure, and leaf health while minimizing the effect of disturbing signals (shadows and soil).

Therefore, it turns out that the spectral resolution impact on the characterization of forest species is more important than that of spatial resolution.

#### 4.2.2. Contribution of Image Time Series

The best obtained results correspond to multi-temporal classifications. Indeed, time series make it possible not only to multiply the number of attributes but also to extract information related to the different phenological stages of the forest. They thus allow for taking into account the phenomenon of changes in leaf color during the seasons and loss of foliage, which differ from one species to another.

#### 4.2.3. Tree Species Assemblage Interpretations

By observing the charts, we notice that the class “Other” is a minority compared to other classes. This is, in fact, related to previous interpretations concerning the low recall values of this class. Another interesting remark concerns the spatial distribution of classes, such that the presence of certain classes is linked to the characteristics of the environment, such as Black Alder, which is often present at the edge of the river. All these data are consistent with a study carried out in the same place [32], which showed that alder trees are specifically located along the river bank and that this riparian forest mainly consists of Pedunculate Oak trees, whereas Maritime Pines and Tauzin Oaks are more often located further from the river. This approach can then be used to test many hypotheses concerning the functioning and dynamics of the forest ecosystem, and conversely, this information could be used as an a priori to enhance the prediction of related classes such as Black Alder.

## 5. Conclusions

This paper focused on tree crown delineation in 2D and 3D and the mapping of forest species by exploiting multi-source spectral and high point-density LiDAR data. Thanks to an automated workflow, several features and data source combinations were tested to assess the contribution of the characteristics of each of them.

The key findings of this study are, first, the contribution of data augmentation by Gaussian noise filtering to overcoming the lack and imbalance of training and testing data. This helped with reaching a 15% increase in validation metrics, as it allowed for the learning of more robust features by the classifier.

Secondly, the study showed the assessment of Sentinel-2 resolution’s impact on forest species classification. In fact, Sentinel-2 high temporal resolution allows for multiplying the number of attributes and getting more relevant information about the different phenological stages, such as leaf color changes during seasons and foliage loss.

Equally important to the temporal resolution contribution, Sentinel-2 spectral richness, especially the red-edge domain, allows for computing numerous attributes that highlight relevant features related to chlorophyll activity, chlorophyll content, internal leaf structure, and leaf health, added to minimizing disturbing signal effects (shadows and soil).

In addition, high-density LiDAR data have great importance in this process. Indeed, it allows for individual tree crown delineation, elevation attribute generation, and pixel co-registration to match Sentinel-2 pixels with accurate positions.

Further work will focus on providing spatial metrics from species patterns and measuring their relationships with biodiversity taxa, especially canopy and sub-strata associations.

## Figures and Tables

**Figure 1 sensors-24-01753-f001:**
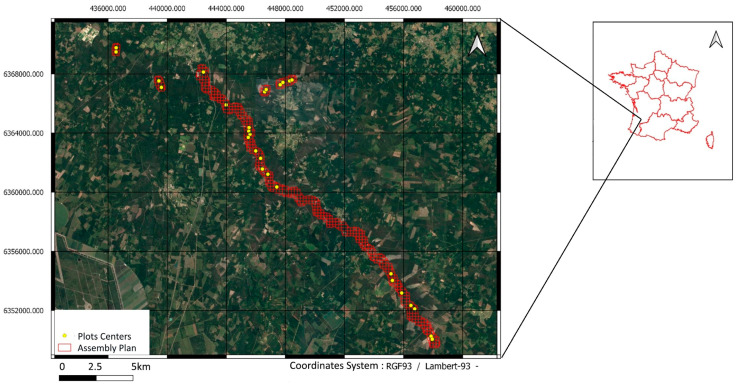
Study area, Ciron Valley, France.

**Figure 2 sensors-24-01753-f002:**
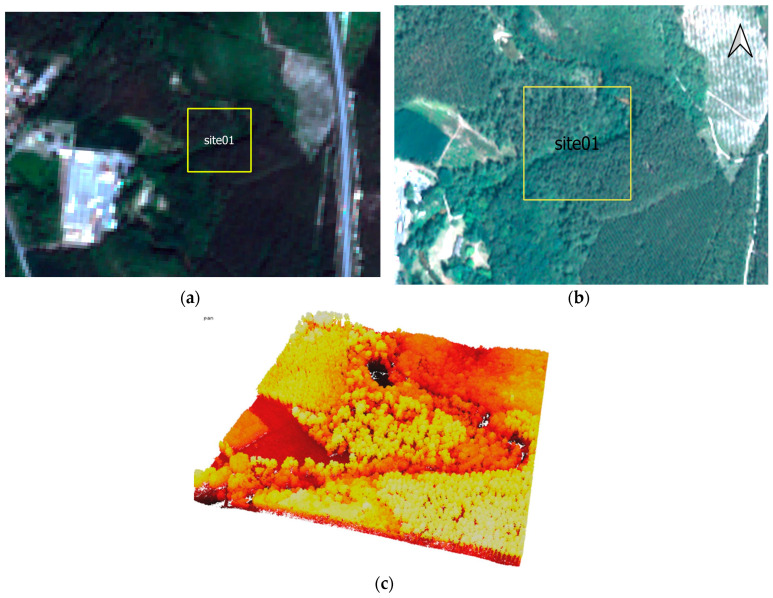
Raw data used covering site 01’s extent: (**a**) multispectral Sentinel-2 image; (**b**) multispectral Pleiades image; (**c**) LiDAR 3D point cloud displayed in black-red-yellow-white color pallet.

**Figure 3 sensors-24-01753-f003:**
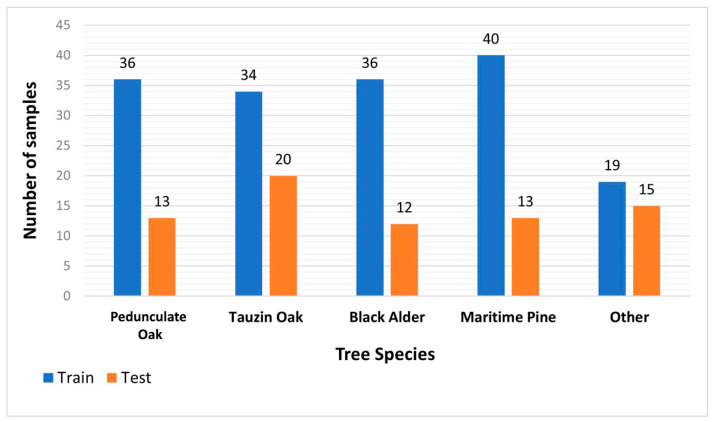
Training and testing samples per class.

**Figure 4 sensors-24-01753-f004:**
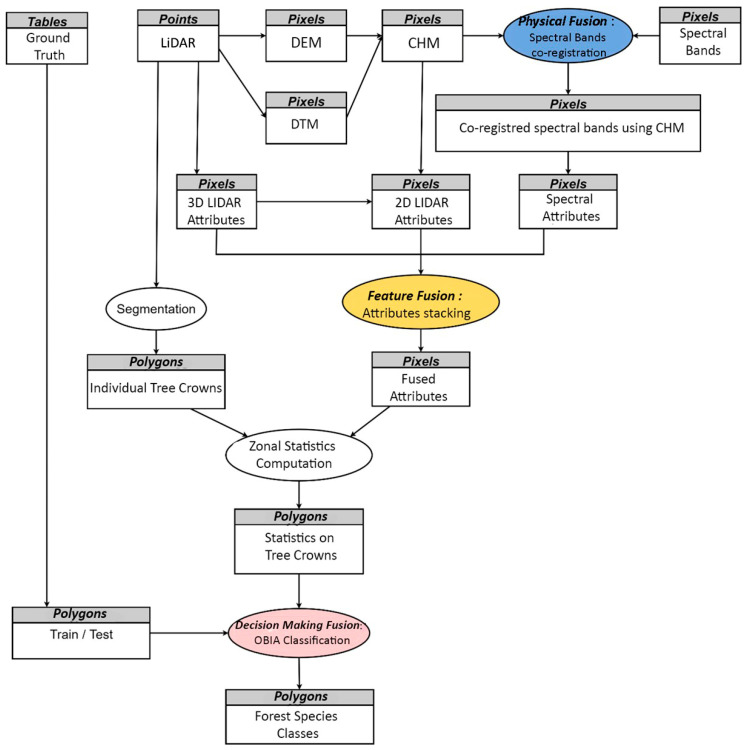
Multi-source data fusion process.

**Figure 5 sensors-24-01753-f005:**
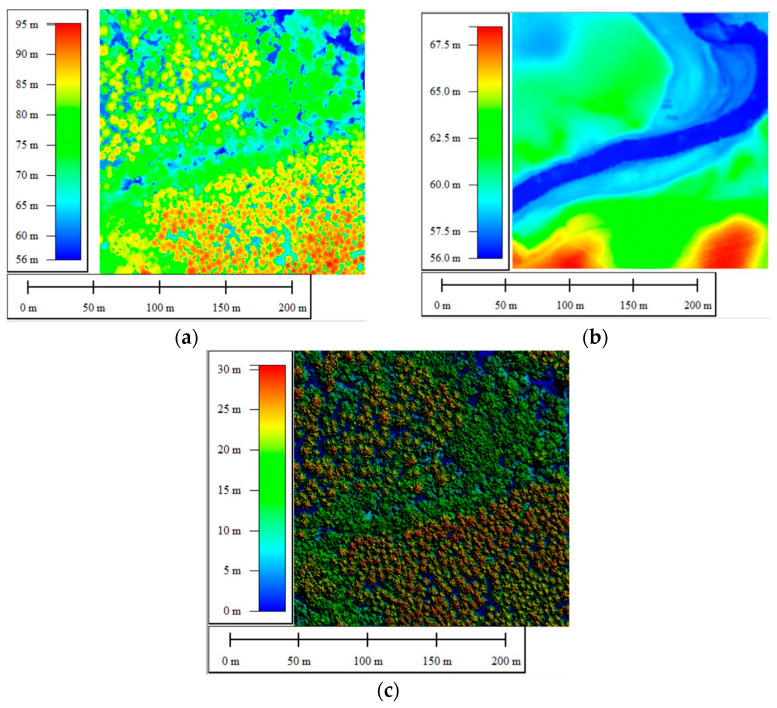
LiDAR 2D attributes: (**a**) density; (**b**) number of echoes; (**c**) elevation range.

**Figure 6 sensors-24-01753-f006:**
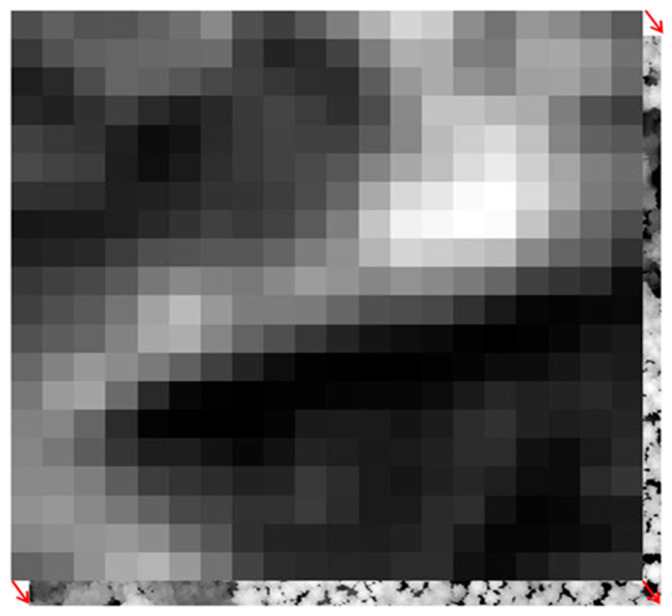
Co-registration of spectral bands to match CHM (base layer) presenting accurate planimetric positions. Red arrows correspond to the applied geometric transformation.

**Figure 7 sensors-24-01753-f007:**
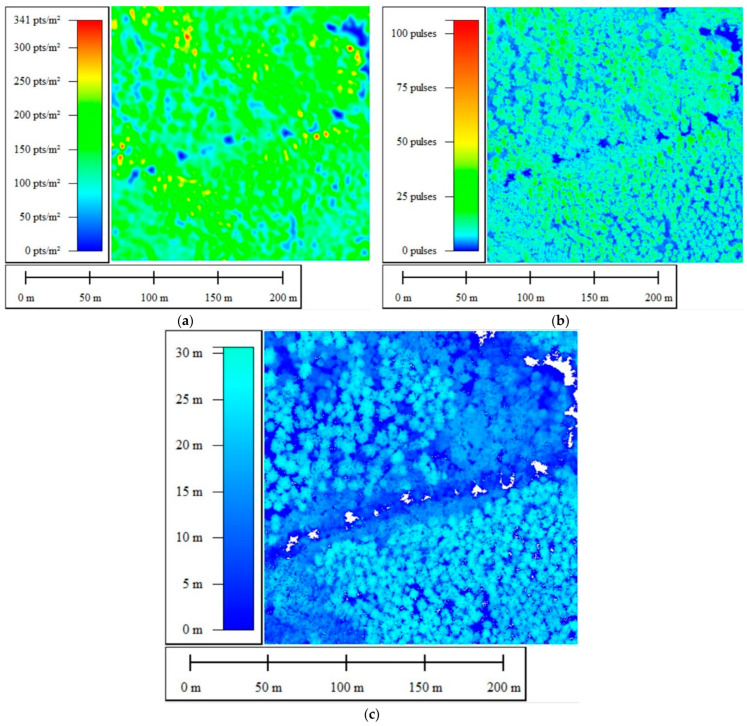
Two-dimensional projection of site 01’s LiDAR attributes: (**a**) density; (**b**) number of echoes; (**c**) elevation range.

**Figure 8 sensors-24-01753-f008:**
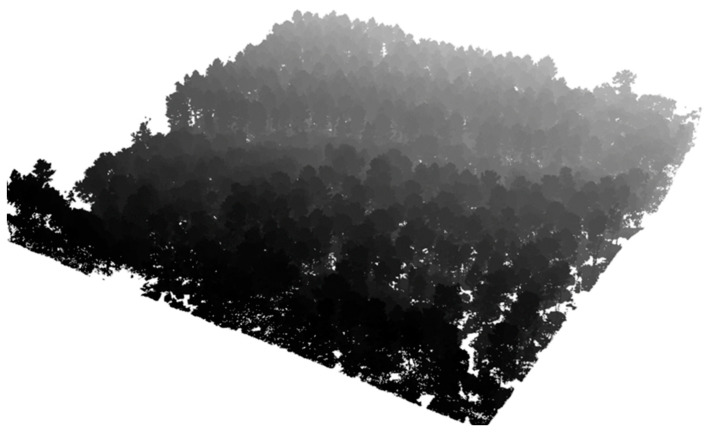
3D Individual tree crown delineation on site 1.

**Figure 9 sensors-24-01753-f009:**
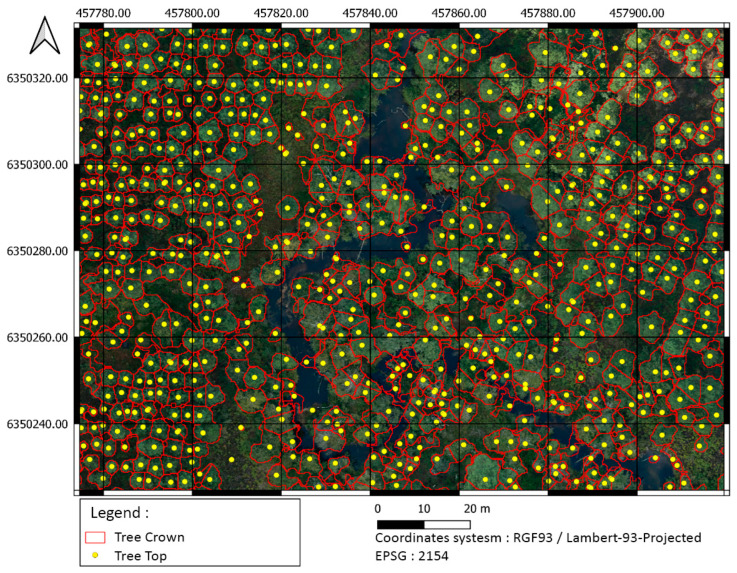
Two-dimensional individual tree crown delineation of site 1 using CHM and PyCrown algorithm. Delimitations are superposed over the VHR Pleiades imagery.

**Figure 10 sensors-24-01753-f010:**
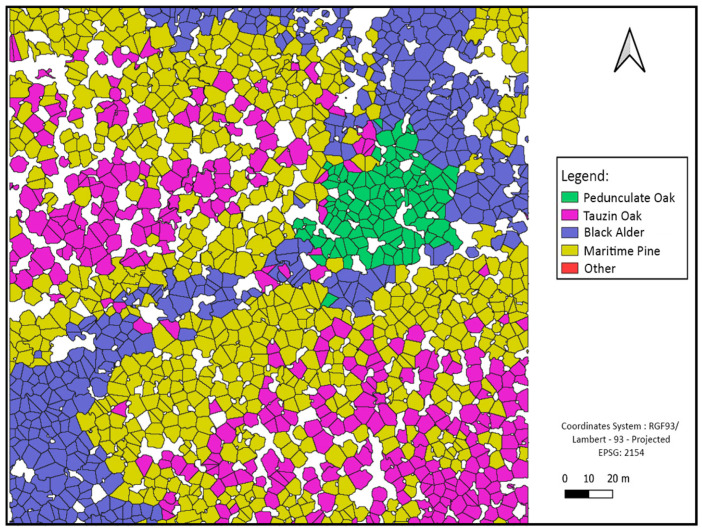
Site 1 forest species classification at tree level using crown delimitations map.

**Figure 11 sensors-24-01753-f011:**
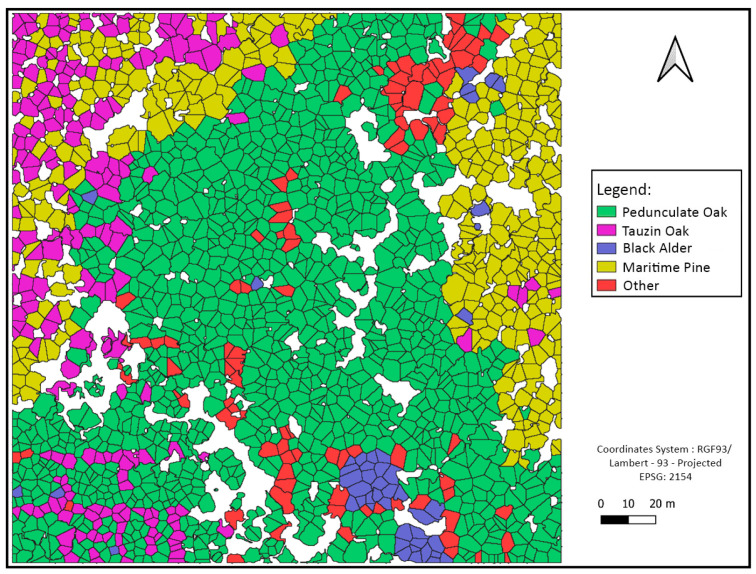
Site 2 forest species classification at tree level using crown delimitations.

**Figure 12 sensors-24-01753-f012:**
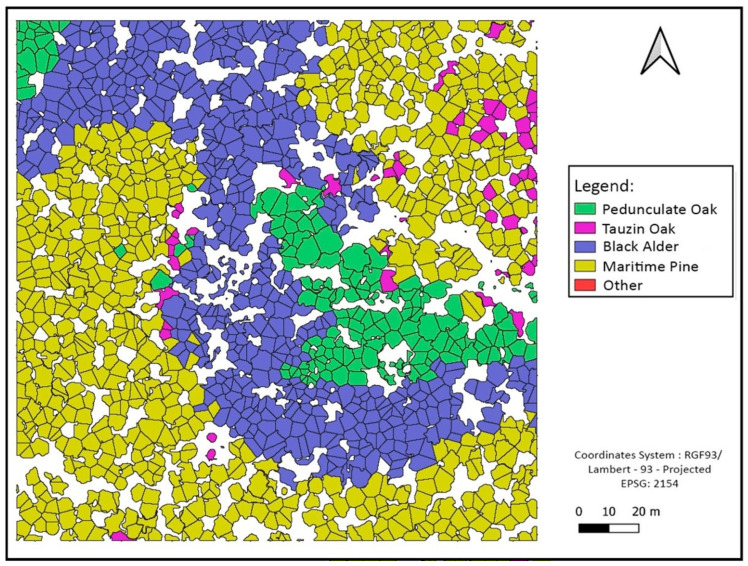
Site 4 forest species classification at tree level using crown delimitations.

**Figure 13 sensors-24-01753-f013:**
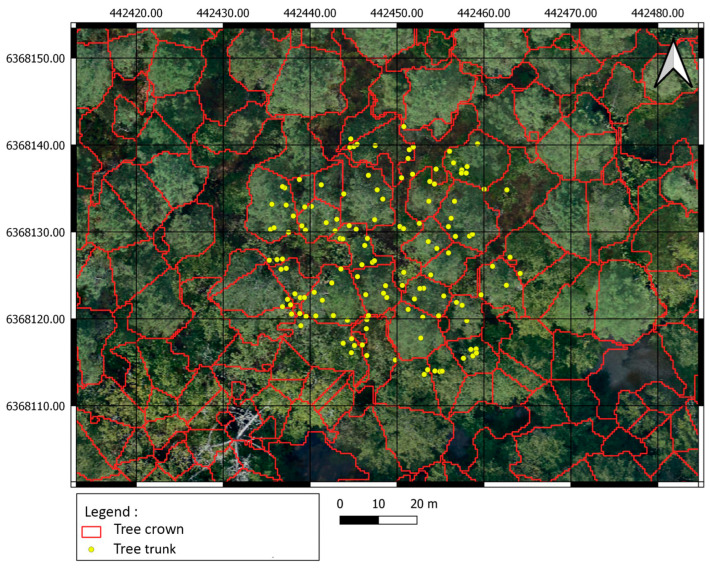
Ground truth data (15 m radius plot) spatial distribution compared to site 01’s extent.

**Table 1 sensors-24-01753-t001:** Sentinel-2 scenes technical information.

Product Tile Reference	Sensor	Radiometric Processing	Dimensions
T30TYQ	Sentinel-2A, Sentinel-2B	Level 2A, top-of-canopy (TOC) reflectance	10,980 × 10,980

**Table 2 sensors-24-01753-t002:** Airborne LiDAR mission details.

Parameters	Description
Date	03/10/2019–04/10/2019
Acquisition system	Laser scan: RIEGL VQ580
	Inertial unit: IXSEA AirINS
	Camera: iXUR 1000–50 mm NIR
Accuracy	Planimetry: 5 cm
	Altimetry: 5 cm
Density	68 pts/m^2^
Projection	RGF 93 Lambert 93 (EPSG: 2154)
Altimetry	IGN69—RAF18

**Table 3 sensors-24-01753-t003:** Summary of average slope and field plot measurements for basal area, stem volume, and total volume for study sites.

	Min	Mean	Max
Slope (degrees)	4.9	8.9	21.0
Basal area (m^2^/ha)	17.2	28.5	47.6
Stem volume (m^3^/ha)	118.7	272.7	475.5
Total volume (m^3^/ha)	135.9	296.2	552.9

**Table 4 sensors-24-01753-t004:** LiDAR attributes used and their utility in tree species classification.

LiDAR Attributes	Definition	Utility
Density	Number of LiDAR points per area unit	Characterizing biodiversity and providing information on the number of strata present in each zone
Number of Echoes	Number of backscatters of the laser pulse	Tree species characterization according to their spatial distribution
Elevation range	Altitude difference between the first and last echo	Tree species discrimination according to their thickness and height

**Table 5 sensors-24-01753-t005:** Spectral indices: formulas and descriptions for Sentinel-2 imagery.

Spectral Index	Formula	Description
NDVI	$\frac{PIR-R}{PIR+R}=\frac{B8-B4}{B8+B4}$	Assesses the importance of biomass and chlorophyll activity.
GRVI1	$\frac{G-R}{G+R}=\frac{B3-B4}{B3+B4}$	- In addition to spring greening, it allows for the detection of autumn coloration, which can be a differentiating factor between hardwoods and softwoods - Robust with misleading signals due to water on the ground surface
CIre	$\frac{REDedge3}{REDedge1}-1=\frac{B7}{B5}-1$	Sensitive to small variations in chlorophyll content, helping differentiation between vegetation classes.
NDVIre3	$\frac{PIR-REDedge1}{PIR+REDedge1}=\frac{B8-B5}{B8+B5}$	Exploits red-edge bands to differentiate between vegetation classes based on the chlorophyll content of the leaves.
Dre2	$\frac{REDedge3-REDedge1}{REDedge3+REDedge1}=\frac{B7-B5}{B7+B5}$	Exploits the red-edge strips to assess the health status of vegetation according to chlorophyll content.
SAVI	$\frac{1.5*(PIR-R)}{PIR+R+0.5}=\frac{1.5*(B8-B4)}{B8+B4+0.5}$	Sensitive to floor’s color and shine, thus minimizing the ground effect.
MSAVI2	$\frac{2*PIR+1-\sqrt{{\left(2*PIR+1\right)}^{2}-8*\left(PIR-R\right)}}{2}=\frac{2*B8+1-\sqrt{{\left(2*B8+1\right)}^{2}-8*\left(B8-B4\right)}}{2}$	Study in vegetation detection in areas with high bare soil composition.

**Table 6 sensors-24-01753-t006:** Python modules used for data fusion process automating.

Python Library	Description
GDAL v1.23.5.1	Translator library for raster and vector geospatial data formats [28].
otbApplication	Python API for Orfeo ToolBox 8.1.0 applications. It is used for image processing and classification [29].
Numpy v1.18	Scientific package for manipulating multidimensional arrays and computing mathematical operations [30].
Whitebox	Package built on WhiteboxTools v1.5.0 [31], an advanced geospatial data analysis platform. It is used to perform common geographical information systems (GIS) analysis operations and LiDAR data processing.
PyCrown	PyCrown [25] is a Python package for identifying treetop positions in a Canopy Height Model (CHM) and delineating individual tree crowns.

**Table 7 sensors-24-01753-t007:** Comparison of classification accuracies using different fusion configurations with and without data augmentation.

Classification		Kappa	OA
Without data augmentation	Sentinel-2 (single date) + LiDAR	0.48	0.59
	Pleiades + LiDAR	0.43	0.55
	Sentinel-2 (time series) + LiDAR	0.51	0.61
	Sentinel-2 (single date) + Pleiades + LiDAR	0.49	0.59
With data augmentation	Sentinel-2 (mono-date) + LiDAR	0.53	0.62
	Pleiades + LiDAR	0.49	0.60
	Sentinel-2 (time series) + LiDAR	**0.66**	**0.69**
	Sentinel-2 (mono-date) + Pleiades + LiDAR	0.58	0.66

**Table 8 sensors-24-01753-t008:** Comparison of precision and recall per species using data augmentation.

	Pedunculate Oak	Tauzin Oak	Black Alder	Maritime Pine	Other
Precision	0.47	0.91	0.54	0.73	0.85
Recall	0.56	0.71	0.89	0.80	0.48

## Data Availability

The data presented in this study are available on request from the corresponding author.

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
