# Peer review of "Fusion of Dense Airborne LiDAR and Multispectral Sentinel-2 and Pleiades Satellite Imagery for Mapping Riparian Forest Species Biodiversity at Tree Level"

_sensors, 2024, doi:10.3390/s24061753_

Round 1
Reviewer 1 Report
Comments and Suggestions for Authors
The article proposes a technology for the joint use of airborn lidar data and satellite multispectral images of various resolutions. Similar studies on aggregation and data fusion are widely known. The novelty in this study is the use of time series of data that allows monitoring the phenological stages helping to discriminate the species. 2D LiDAR attributes with vegetation indexes were obtained and combined at the object level. At the decision-making level, LiDAR data was used to determine the three-dimensional boundaries of the crown of trees, providing unique trees or groups of trees that are used as the basis for species classification. In general, there are no comments on the article, it can be published in this form.
Author Response
Dear reviewer,
Thank you for your review.
Please see the attachment for specific responses.

Reviewer 2 Report
Comments and Suggestions for Authors
The paper employs three distinct remote sensing datasets for species identification at the tree canopy level, assessing the contribution of each dataset to tree classification. This approach is significant for forest resource assessment and environmental protection. However, I have several concerns and suggestions:
a. Study Area and Data: The paper provides statistical information about the study area and data but lacks accompanying visual aids. I noted a reference to Figure 1 in line 62, yet the image is absent. Additionally, raw visualizations of the data used should be included.
b. Methodology: The methodology comprises three primary steps: 1. LiDAR and multispectral image co-registration, 2. Feature fusion, and 3. Decision-making fusion. Each step lacks detailed description. For instance, the registration process is only depicted in Figure 4 without clear explanation of the method used or identification of corresponding points. Similarly, Figure 4 does not clearly indicate which image represents the CHM model. Figures 9 and 10 suffer from the same issue. Regarding decision-level fusion, line 192 mentions five feature combinations, but only four are presented in Table 6, leading to confusion. The use of a random forest classifier is mentioned, but there is no detailed description of its parameter settings.
c. Introduction: While the article reviews some previous research, it fails to articulate how this study differs from prior work, contributing to its lack of innovativeness.
Comments on the Quality of English LanguageSome words have spelling errors, please correct.
Author Response

(The authors gave the same response as above.)

Reviewer 3 Report
Comments and Suggestions for Authors
This manuscript developed an automatic scheme to map riparian forest species at tree level using a fusion of Airborne LiDAR data and Sentinel-2 and Pleiades imagery. The most attractive idea is data fusion at feature-level and decision level. However, the manuscript could be improved in several aspects before it can be accepted for publication.
1) Abstract: the abstract lacks the description of data fusion and classification methods, and it is hard to identify the innovation ideas of the paper. It is suggested to add relative contents.
2) Introduction: The research review is incomplete and provides few details about the method developments on mapping riparian forest species. Please make supplements on more references and summaries.
3) 2.3 Methodology: There is few descriptions about the automated workflow to map riparian forest species biodiversity at tree level, and now the workflow map cannot show the automatic process.
4) 3.2 classification: The mapping results lack the comparisons between the developed method and existing methods. Moreover, the ground truth data was not mentioned in this section.
Author Response

(The authors gave the same response as above.)

Reviewer 4 Report
Comments and Suggestions for Authors
The overall structure of the paper is complete and the research is carried out by the real-measured data. On the whole, it is a qualified academic paper, but there still remain several problems:
1. It is suggested to refer to more relevant literature in the last five years, in order to clarify the latest frontier research direction, and expand the content of the introduction part;
2. There are many problems in the writing of some formulas, and the format and content need to be modified to ensure accuracy;
3. The superiority of the proposed method is insufficient, and relevant data need to be expanded to verify the superiority of the proposed algorithm;
4. The conclusion of this paper is too redundant, so it is suggested to compress the content in short form and write briefly;
5.The quality and size of some images need to be adjusted.
Comments on the Quality of English LanguageThe overall writing quality of this paper is not very outstanding and it needs to be polished up.
Author Response

(The authors gave the same response as above.)

Round 2
Reviewer 2 Report
Comments and Suggestions for Authors
The quality of this article has improved after the first revision. It is suggested to remove the words 'riparian' from the title, as it seems to have little relevance to the overall content of the article.
Comments on the Quality of English LanguageBe mindful of some spelling mistakes.
Author Response
Thank you for your review, please view the attachment.

Reviewer 4 Report
Comments and Suggestions for Authors
The revised version of this paper basically solves the related problems mentioned before and is a qualified academic paper.
Comments on the Quality of English LanguageThe revised version meets the language requirements of the academic papers published on Sensors.
Author Response

(The authors gave the same response as above.)
